# Circadian Analysis of the Mouse Cerebellum Proteome

**DOI:** 10.3390/ijms20081852

**Published:** 2019-04-15

**Authors:** Marine Plumel, Stéphanie Dumont, Pauline Maes, Cristina Sandu, Marie-Paule Felder-Schmittbuhl, Etienne Challet, Fabrice Bertile

**Affiliations:** 1Institut Pluridisciplinaire Hubert Curien, LSMBO, Centre National de la Recherche Scientifique (CNRS), Université de Strasbourg, 67087 Strasbourg, France; marine.plumel@gmail.com (M.P.); Pauline.Maes@u-bourgogne.fr (P.M.); 2Institute of Cellular and Integrative Neurosciences, CNRS, Université de Strasbourg, 67000 Strasbourg, France; dumonts@inci-cnrs.unistra.fr (S.D.); sandu@inci-cnrs.unistra.fr (C.S.); feldermp@inci-cnrs.unistra.fr (M.-P.F.-S.)

**Keywords:** circadian rhythm, mouse, cerebellum, 2D-DIGE/MS

## Abstract

The cerebellum contains a circadian clock, generating internal temporal signals. The daily oscillations of cerebellar proteins were investigated in mice using a large-scale two-dimensional difference in gel electrophoresis (2D-DIGE). Analysis of 2D-DIGE gels highlighted the rhythmic variation in the intensity of 27/588 protein spots (5%) over 24 h based on cosinor regression. Notably, the rhythmic expression of most abundant cerebellar proteins was clustered in two main phases (i.e., midday and midnight), leading to bimodal distribution. Only six proteins identified here to be rhythmic in the cerebellum are also known to oscillate in the suprachiasmatic nuclei, including two proteins involved in the synapse activity (Synapsin 2 [SYN2] and vesicle-fusing ATPase [NSF]), two others participating in carbohydrate metabolism (triosephosphate isomerase (TPI1] and alpha-enolase [ENO1]), Glutamine synthetase (GLUL), as well as Tubulin alpha (TUBA4A). Most oscillating cerebellar proteins were not previously identified in circadian proteomic analyses of any tissue. Strikingly, the daily accumulation of mitochondrial proteins was clustered to the mid-resting phase, as previously observed for distinct mitochondrial proteins in the liver. Moreover, a number of rhythmic proteins, such as SYN2, NSF and TPI1, were associated with non-rhythmic mRNAs, indicating widespread post-transcriptional control in cerebellar oscillations. Thus, this study highlights extensive rhythmic aspects of the cerebellar proteome.

## 1. Introduction

The circadian system controls most aspects of behavior and metabolism on a 24 h time-scale. More specifically, circadian rhythmicity regulates the timing of physiological mechanisms via daily organisation (e.g., anticipation of predictable events) and partitioning (e.g., temporal segregation of incompatible functions) at the cellular, tissue and organism levels. In mammals, the circadian system is a multi-oscillatory network comprising a master clock in the suprachiasmatic nuclei of the hypothalamus and a number of secondary clocks in the brain and peripheral organs, such as the liver. An internal cyclic timing is achieved by the master clock that, like a conductor, conveys timing signals via neural and humoral pathways to secondary clocks throughout the body [1,2].

The molecular mechanisms of the circadian clocks rely on auto-regulatory transcriptional and translational feedback loops that generate a rhythmic regulation of specific genes, called clock genes. A positive loop involves two transcriptional factors, CLOCK and BMAL1, that activate the transcription of E-boxes containing clock genes such as *Period* (*Per1,2,3*) and *Cryptochrome* (*Cry1,2*). A negative loop involves PER and CRY proteins that repress transactivation mediated by CLOCK-BMAL1 [3]. In addition to these main loops, the nuclear receptors RORα,β,γ and REV-ERBα,β compete to respectively stimulate and repress the expression of *Bmal1* and *Clock* [4,5,6]. A number of clock proteins undergo post-translational modifications, such as phosphorylation, acetylation, ubiquitination, and sumoylation that modulate characteristics (i.e., phase, amplitude or period) of circadian oscillations [7,8]. The positive circadian loop mediated by CLOCK-BMAL1 dimers controls the rhythmic transcription of clock-controlled genes, whose proteins provide temporal cues within and eventually out of the clock cells [3].

Among the numerous cerebral regions outside the master clock that participate in the 24 h rhythmic function of the brain [2], the cerebellum harbours a circadian clock, as shown by self-sustained oscillations when isolated in vitro [9]. Daily patterns of clock gene expression in the cerebellum and other secondary clocks such as the liver show a temporal organization of the clockwork very close to that in the suprachiasmatic clock, as described above [9,10]. In terms of phasing of the whole clockwork, however, circadian oscillations of clock genes and proteins in the mouse cerebellum are rather similar to those in the murine liver, the oscillations in both structures being phase-delayed by about 6 h as compared to those in the master clock. For instance, *Per1* and *Dbp* mRNA profiles peak around dusk (corresponding to activity onset in mice) in both the cerebellum and liver, while these profiles display peaks around midday (mid-resting period) in the suprachiasmatic nuclei (i.e., 6 h earlier) [11,12,13]. The cerebellum is classically involved, among others, in coordination and accurate timing of motor tasks [14]. Although the functional significance of the circadian clock in the cerebellum is not fully understood yet, our previous work has involved the cerebellar clock in the daily anticipation of mealtime [9].

The circadian proteome in the liver has been deeply investigated, leading to new views on the regulating modalities of the circadian system. Notably, many proteins are rhythmically expressed with constitutive mRNA and vice versa [15,16]. In the brain, circadian proteomic analyses have been mostly focused on the master clock [17,18]. These studies have highlighted several outcomes, including synaptic vesicle cycling and rhythmicity of a large set of mitochondrial proteins. Whether these features are specific to the master clock or brain clocks deserves further investigation. For that purpose, cerebral clocks in other structures than the suprachiasmatic nuclei should be studied. Therefore, the present work aimed at gaining more insights into how the circadian clock in the cerebellum affects its function on a daily basis. To this end, we set up a proteomic approach based on visualisation of the proteome of mouse cerebellum using the two-dimensional difference in gel electrophoresis (2D-DIGE) and the identification of proteins of interest by mass spectrometry (MS).

## 2. Results

For each time-point, 2D-DIGE was based on the analysis of *n* = 4–6 C57BL/6J individual cerebellar samples, i.e., no mix of samples from a same time-point were made. This led to the detection of 588 protein spots (for an example of 2D gel image, see Appendix A), among which 26 (i.e., 5% from all spots) were found to be rhythmic according to cosinor procedure, with an additional one being very close to significance (see below). One hundred fourteen proteins were identified in these 27 spots (see details on protein annotations and statistical results in Appendix A), of which forty-seven most abundant proteins were selected (see Methods) and are hereafter referred to as rhythmic proteins showing daily variations in their abundance.

Functional annotation enrichment analysis allowed the classification of these rhythmic proteins into main broad functions, including synapse and trafficking, carbohydrate metabolism, protein metabolism and mitochondrial function (Table 1, Figure 1).

Other main broad functions highlighted here included energy metabolism and cytoskeleton (Figure 1). Five proteins (about 10%, namely Cytochrome b-c1 complex subunit 1 [UQCRC1], ATP synthase subunit β [ATP5B], Vesicle-fusing ATPase [NSF], D-Dopachrome decarboxylase [DDT] and Carbonic anhydrase 2 [CA2]) were rhythmically expressed in two different protein spots on 2D gels (Table 1), likely representing isoforms of the same protein due to post-transcriptional modifications.

There was a temporal segregation in the rhythmic expression of cerebellar proteins, leading to a clear bimodal timing (Figure 2; Table 1). More precisely, half of them (22/47) were peaking daily close to midnight (i.e., Zeitgeber time (ZT) 18 corresponding to the middle of the active period), and about two-fifths (18/47) were peaking around midday (i.e., ZT6 corresponding to the middle of the resting period). The few remaining most abundant proteins (7/47) were expressed at higher levels in late afternoon (ZT9-ZT12). Remarkably, no peak of rhythmic proteins was detected during the first half of nighttime and during a 9 h temporal window spanning from late night to early morning (i.e., from ZT19 to ZT4).

Importantly, 8/47 proteins (i.e., 17%) were mitochondrial and the majority of them (5/8) displayed a peak of abundance clustered to midday (mid-resting phase; Table 1). This was notably the case for proteins of the respiratory chain, including NADH dehydrogenase [ubiquinone] iron-sulfur protein 2 (NDUFS2), UQCRC1 and ATP5B. On the reverse, ATP synthase subunit α (ATP5A1) peaked in the middle of the dark active phase.

Six enzymes involved in carbohydrate metabolism were rhythmically expressed over 24 h (6/47). Two of them peaked around the middle of the rest phase (Glucose 6-phosphate 1-dehydrogenase [G6PDX] and pyruvate dehydrogenase E1 [PDHB], which is located in the mitochondria), another one was more highly expressed in late daytime (α-enolase [ENO1]) and the four others were mostly expressed in the middle of the active phase (Triosephosphate isomerase [TPI1], Phosphoglycerate mutase 1 [PGAM1], Transaldolase [TALDO1] and Phosphoglycerate kinase 1 [PGK1]).

The rhythmic proteome also contained a molecular chaperone, Heat shock protein (HSP) 90-α (HSP90AA1), and a proteolytic-related factor, the Proteasome subunit α type-1 (PSMA1), which abundances were higher during the resting phase.

Several protein synthesis-related factors were also found to be rhythmic, with a peak phase at midday (Spliceosome RNA helicase [DDX39B]) or in the active phase (Eukaryotic translation initiation factor 4H [EIF4H] and Elongation factor 2 [EEF2]).

Visualisation of the daily expression of proteins on metabolic maps highlights that rhythmic protein expression levels are globally not coordinated within and between metabolic pathways (Figure 3).

Finally, 16/47 rhythmic proteins (i.e., 34%) were involved in the synapse activity, trafficking and cytoskeleton (Figure 2), including NSF and β-synuclein (SNCB) both peaking around midnight, and Synapsin 2 (SYN2) peaking in contrast close to midday (Figure 3). The respective nocturnal and diurnal rises in NSF and SYN2 protein contents in the cerebellum were confirmed by Western blots (Figure 4; Table 1).

To investigate whether NSF, SYN2, SNCB, and TPI1 protein rhythms were controlled by rhythmic transcription, we investigated the circadian patterns of corresponding mRNAs in the cerebella of wild-type (WT) mice. The circadian expression of the clock gene *Bmal1* in the mouse cerebellum peaked in early morning, as already reported in nocturnal rodents [10,19]. In contrast, none of mRNA levels coding for NSF, SYN2, SNCB, and TPI1 were found to be rhythmically expressed in WT mice, suggesting that rhythmicity of these proteins relies on translational or post-translational mechanisms downstream of transcriptional loops (Figure 5). Moreover, as expected, the circadian expression of *Bmal1* was lost in mice with defective clocks (i.e., *Per1^-/-^/Per2^Brdm1^* double mutant mice). Arrhythmicity of mRNAs coding for NSF, SYN2, SNCB, and TPI1 was confirmed in mutant mice. Next, we used expression of the clock-controlled gene *Fabp7* as a positive control because this gene is rhythmically transcribed in the brain [20,21]. Accordingly, mRNA expression of *Fabp7* was rhythmic (circadian) and arrhythmic in the cerebella of WT and *Per1^-/-^/Per2^Brdm1^* mice, respectively (Figure 5). Of note, the daily expression of FABP7 protein (spot N°42) in the cerebellum of WT mice was close to our criteria of circadian rhythmicity as only its *p* value for amplitude was just above the threshold of significance (*p* = 0.06) (Table 1, Appendix A).

## 3. Discussion

The aim of this study was to provide more insights into the circadian clock of the cerebellum, a hindbrain structure known for its critical role in motor coordination. Hence, we analysed the circadian landscape of cerebellar proteome to determine which rhythmic proteins and key biological pathways are enriched.

Among the 27 spots containing rhythmic proteins in the cerebellum of C57BL6J mice, 47 identified proteins displayed daily variations. Amongst these proteins, there were two main peaks of abundance, centered either in the middle of the rest (midday) or active phases (midnight). Of note, such bimodal clustering of rhythmic cerebellar proteins over 24 h differs from the suprachiasmatic nuclei in which the peaks of protein abundance are more widely distributed in phase across the 24 h [17,18]. Thus, the daily distribution of rhythmic proteins in the cerebellum did not match the lagged temporal organization of its clockwork when compared to the suprachiasmatic nuclei. These findings therefore suggest complex temporal regulations that differ between cerebral structures. In the peripheral clock of the liver, circadian profiles of protein expression are also differently distributed over 24 h because in that case, they fall into three distinct phase clusters, the main one peaking at night and two minor ones peaking either around midday or at the day-to-night transition [16].

Clock proteins were most likely not expressed at levels high enough to be detected by 2D-DIGE analyses. This was also the case in the previous circadian studies using the same proteomic method in the mouse suprachiasmatic nuclei and even in the liver, or in the rat pineal gland [16,17,22]. Nevertheless, the present study shows that the clock gene *Bmal1* and the clock-controlled gene *Fabp7* are rhythmically expressed in the cerebellum of wild-type mice, while these rhythms are totally abolished in mice with defective clocks (i.e., *Per1^-/-^/Per2^Brdm1^* double mutant mice). Such observation confirms that the cerebellum activity is controlled, at least in part, by a circadian clock.

### 3.1. Rhythmic Proteins in Mitochondria of Cerebellar Cells

Mitochondrial proteome in hepatocytes is highly rhythmic throughout the day with about 38% of mitochondrial proteins oscillating on a daily basis [23]. A typical feature of these oscillations is their timing, with daily peaks occurring mostly in the morning (i.e., early resting phase in nocturnal mice). Such oscillations in the liver have been shown to be regulated by the clock proteins, PER1 and PER2 [23]. In the master clock of the suprachiasmatic nuclei, rhythmic mitochondrial proteins were also found enriched (26% of oscillating proteins), but in that case mostly at night (i.e., throughout the active phase of the mice) [18]. Remarkably, even if the oscillating mitochondrial proteins are distinct, we found that rhythmic mitochondrial proteins in the cerebellum were especially abundant during the (early) light/resting phase as in the liver, thus suggesting comparable regulation of daily timing in mitochondria within multiple secondary clocks throughout the body. Such temporal regulation with more abundant mitochondrial proteins during daytime likely explains the higher ATP levels during daytime in the mouse liver [24], the hippocampus [25], and possibly in the cerebellum as well. An intriguing observation was that subunits from a same oxidative phosphorylation (OXPHOS) complex did not peak in a coordinated fashion in the mice cerebellum (see ATP5A1 and ATP5B belonging to the complex V). A similar observation has been made in the suprachiasmatic nuclei for subunits of the OXPHOS complex I [18]. Such differences in the expression timing of subunits of the same functional complexes may help to finely tune mitochondrial respiration and ultimately the overall cellular energetics to meet temporal requirements in cell functioning.

### 3.2. Rhythmic Proteins in the Cytosol of Cerebellar Cells

It is clear that the circadian spectrum of oscillating cytosolic proteins differs largely from one circadian clock to another (i.e., they are brain region- and structure-specific) because the vast majority of oscillating proteins in the cerebellum were not identified as rhythmic in the retina, the suprachiasmatic nuclei or the liver [16,17,26]. Nevertheless, Alpha-enolase (ENO1) is among the rare cytosolic proteins showing rhythmic expression in various circadian clocks, including the master clock (suprachiasmatic nuclei) [17] and two secondary clocks, the liver [16] and the cerebellum (this study). Triosephosphate isomerase (TPI1) was also found rhythmic in the two cerebral clocks investigated so far, namely, suprachiasmatic nuclei [17] and cerebellum (this study). However, the peak of daily abundance of ENO1 or TPI1 being not provided in the study focused on the suprachiasmatic nuclei [17], we cannot compare their respective phases with those in the cerebellum. In addition to ENO1 and TPI1, other rhythmic proteins involved in carbohydrate metabolism have been identified, as well here in the cerebellum as in the suprachiasmatic nucleus [18] and the liver [16] of mice, but also in fungi like *Neurospora crassa* [27]. This confirms that the temporal expression of carbohydrate metabolism-related factors has likely evolved as a key mechanism in any living cell or organism.

Metabolically speaking, other proteins were found among the rhythmic most abundant proteins in the cerebellum. In particular, Sirtuin 2 (SIRT2), a member of the sirtuin family that comprises NAD^+^-dependent histone deacetylases, is mainly present in the cytoplasm, where it deacetylates a number of substrates, including α-tubulin, histone H4 and p53 [28]. It also exhibits other functions in various physiological processes, including energy and substrate (e.g., carbohydrates) metabolism [29]. Its higher expression levels in the cerebellum during the active phase, i.e., at the same time when four of six proteins of the metabolism of carbohydrates also peak, could thus be linked to energy metabolism and the changes in nutrient availability when mice are eating.

In both the suprachiasmatic nuclei and liver, expression of a heat shock protein (HSPA9A) has been found to be affected by time of day [16,17]. HSPA9A, also termed glucose-responsive protein 75 (GRP75), is a key protein involved in the crosstalk between endoplasmic reticulum and mitochondria, notably in neuronal cells [30]. Here in the cerebellum, another HSP, namely Heat shock protein HSP90-alpha (HSP90AA1), displays daily changes of abundance peaking at midday. HSP90aa1 is the stress inducible isoform of the molecular chaperone HSP90, being involved in the structural maintenance of specific target proteins [31]. No clear day-night variation of this protein, known to interact with glucocorticoid receptors, was found in the hypothalamus and hippocampus [32], indicating a specific rhythmic feature of the cerebellum in that respect. Spliceosome RNA helicase (DDX39B), a protein known to stabilize pre-ribosomal RNA [33], was found rhythmic in the cerebellum, with a peak of abundance at midday. Hence, concerning the overall process of protein synthesis, the factors promoting stability (HSP90AA1 and DDX39B) are antiphase to proteins involved in translation initiation (EIF4H [34]) and elongation (EEF2 [35]), which may represent a mechanism by which rhythmicity of proteins is regulated in cerebellar cells.

Carbonic anhydrases (CA) are enzymes playing a buffering role in the brain that are enriched during anoxic stress [36]. A close connection between expression of CA2 and the circadian clock has been revealed by the longer free-running period in mice deficient for CA2 [37]. CA2-deficient mice also exhibit low locomotor activity compared to either heterozygous or WT litter mates [38]. In the cerebellum, both CA1 and CA2 show a daily peak of abundance during the active period in the current study. Altogether, these data argue for a role of CA2 in the cerebellar control of motor movements.

A number of rhythmic proteins in the cerebellum are related to the synapse activity and trafficking. Based on self-sustained circadian oscillations of clock genes ex vivo (e.g., *Per2*), both the suprachiasmatic nuclei and the cerebellum contain circadian clocks [9]. Rhythmic electrical activity of suprachiasmatic cells peaking during daytime is thought to distribute circadian signals from the master clock [39,40]. By contrast, in the cerebellum, circadian signals are not transduced into a strong daily rhythm of firing rate [41]. Still, parvalbumin (PVALB), a calcium-binding protein expressed in the Purkinje cells and cerebellar interneurons that plays a role in the rhythmic firing rate of Purkinje cells [42], displays a peak of abundance around midnight in the cerebellum (this study). In addition, two important proteins for the synapse activity and trafficking, i.e., Synapsin 2 (SYN2) [43] and Vesicle-fusing ATPase (NSF) [44], were found to be rhythmically present in both the suprachiasmatic nuclei [17] and the cerebellum (this study). However, while the suprachiasmatic nuclei are enriched concomitantly in SYN2 and NSF at the onset of the active phase (i.e., CT12, that is, after the circadian peak in neuronal firing rate), the rhythmic expression of cerebellar SYN2 and NSF is opposite in phase (mid-resting and mid-active phase, respectively). The lack of rhythmic transcription of both NSF and SYN2 in the cerebellum of WT mice is in accordance with findings in the suprachiasmatic nuclei [17]. This indicates that rhythmic expression of NSF and SYN2 proteins are not under transcriptional control in the brain, and we extend this conclusion to β-synuclein and other cerebellar proteins such as TPI1.

## 4. Materials and Methods

### 4.1. Ethical Statement

The experiments were performed in accordance with the NIH Guide for the Care and Use of Laboratory Animals (1996) and the French National Law (implementing the European Union Directive 2010/63/EU) and approved by the Regional Ethical Committee of Strasbourg for Animal Experimentation (CREMEAS) and French Ministry of Higher Education and Research (APAFIS #2533-2015110215078867 v1).

### 4.2. Animals and Study Design

Male C57BL/6J, 3-month-old mice (Janvier labs, Le Genest-Saint-Isle, France) were housed in an animal care facility (Chronobiotron platform, UMS 3415, CNRS and University of Strasbourg) at 22 ± 1 °C under a 12:12 h light/dark cycle (lights on at 07:00 a.m.). Mice had *ad libitum* access to food (Rodent chow, diet 105, Scientific Animal Food and Engineering, Augy, France) and UV-treated tap water. On the day of sampling, food was withdrawn during daytime (resting period) and given back at night (active period). Four groups of mice were sequentially injected intraperitoneally with a lethal dose of pentobarbital (200 mg kg^−1^; CEVA, Libourne, France) every 6 h, starting 5 h after lights on (i.e., ZT (Zeitgeber) 5; *n* = 5), followed by 11 h after lights on (i.e., ZT11; *n* = 6), 6 h later at night (i.e., ZT17; *n* = 5) and finally at ZT23 (i.e., 1 h before lights on; *n* = 5). Cerebellum was harvested and immediately frozen in liquid nitrogen and stored at −80 °C. One sample at ZT5 was lost, leading to a final *n* = 4.

Male 2-month-old *Per1^-/-^/Per2^Brdm1^* double mutant mice and WT C57BL/6 X 129Sv individuals from the same strain bred in Chronobiotron were used for mRNA analysis. These strains were kindly provided by Prof. Urs Albrecht (University of Fribourg, Fribourg, Switzerland). The loss-of-function *Per1* mutation (*Per1^-/-^*) and the *Per2* mutation (*Per2^Brdm1^*) are described in [45,46], respectively. On the day of sampling, mice were transferred to constant darkness. Following euthanasia by CO_2_ and cervical dislocation performed in complete dark using night vision goggles ATN NVG-7 (American Technologies Network Corp, San Francisco, CA, USA), cerebella were frozen on dry ice every 6 h, starting 4 h after the projected time of lights on (i.e., circadian time (CT) 4; *n* = 3 per genotype), followed by CT10 (*n* = 5 per genotype), CT16 (*n* = 3 per genotype) and CT22 (*n* = 5 per genotype). Note that the timing of cerebella sampling in these control and mutant mice was phase-advanced by 1 h, as compared to the previous experiment in wild-type C57BL/6J mice. Because we assessed circadian phase by non-linear sine regression (cosinor) over the four time-points (see statistical analysis for details) and we did not compare data between the two experiments, the 1 h difference in sampling does not affect the validity of our results.

### 4.3. Cerebellum 2D-DIGE-MS Analysis

Unless otherwise specified, all reagents and chemicals were purchased from Sigma Diagnostics (St. Louis, MO, USA). Protein extraction was performed from half of each cerebellum. Briefly, samples were sliced thinly on ice using scissors, and incubated for 1 h 30 min under agitation in a denaturing buffer composed of 8 M urea, 2 M thiourea, 4% Chaps, 1% DTT, 0.5% Triton X-100, 0,005% TLCK, and protease inhibitors at 0.02 to 2 mM. After sonication on ice (10–30 s, 135 W), followed by centrifugation (5 min, 12,000× *g*, 4 °C) to remove cell debris, proteins were TCA-precipitated and pelleted by centrifugation (15 min, 14,000× *g*, 4 °C). Proteins were then vacuum-dried using a Speedvac (ThermoFisher Scientific; Rockford, IL, USA), dissolved in a solution composed of 7 M Urea, 2 M thiourea, 30 mM Tris (pH 8.5) and 4% Chaps. Homogeneization was afterwards completed by sonication on ice (10 s, 135 W). Protein concentrations were determined using the Bio-Rad Protein Assay (Bio-Rad, Hercules, CA, USA). Prior to quantitative DIGE analyses, similarity of protein profiles between all samples was checked on 10% SDS-PAGE acrylamide gels (10 μg loaded; 50 V for 90 min and then 100 V to complete migration) using Coomassie blue (Fluka, Buchs, Switzerland) staining.

Protein samples were labelled using a CyDye DIGE Fluor Minimal Dye Labeling Kit (GE HealthCare, Uppsala, Sweden). CyDyes were first reconstituted in anhydrous N,N-dimethylformamide, then 400 pmol of Cy3 and Cy5 were used to randomly label 50 μg of protein samples from the different groups, and 7.2 nmol of Cy2 were used to label a mixture of all the samples (equal amounts from each protein extracts) that was used as an internal standard. After incubation in the dark for 30 min on ice, protein labelling was quenched by addition of 10 mM lysine and incubation in the dark for 10 min on ice.

Prior to 2D gel electrophoresis, two labelled samples (50 µg each) and 50 µg of labelled internal standard were pooled and the multiplexing of samples from the different groups was randomized to avoid any bias. Briefly, Cy2, Cy3 and Cy5-labelled protein samples were mixed and diluted with 7 M urea, 2 M thio-urea, 2% Chaps, 2% DTT, 2% ampholytes (SERVALYT Carrier Ampholytes 3-10, Serva, Heidelberg, Germany) and a trace of bromophenol blue to a total volume of 400 μL. Proteins were then loaded onto 24 cm pH3-10 non-linear immobilized pH gradient strips (IPG Ready strip, Bio-Rad, Hercules, CA, USA), and left in the dark for passive rehydration over 1 h. Active rehydration was then performed overnight at 50 V using a Protean IEF cell (Bio-Rad), and subsequent isoelectric focusing (IEF) was performed using voltage gradient steps (from 0 to 200 V in 1 h, from 200 to 1000 V in 4 h, from 1000 to 5000 V in 16 h, then 5000 V for 9 h), with a total focusing time of 95,500 Vh. Focused proteins were then reduced and alkylated by incubation of IPG strips in DTT and iodoacetamide equilibration buffers Serva) for 15 min each. IPG strips were then placed into the slot of 12.5% polyacrylamide SDS-PAGE precast gels (2D HPE large gel NF 12.5% kits), and electrophoresis was carried out using the HPE Flap Top Tower (Gel company, San Francisco, CA, USA), applying 7 mA per gel for 30 min, 13 mA for 30 min, 20 mA for 10 min, 40 mA for 3 h 50 followed by 45 mA for 40 min. 

After washing steps with water, gel images were acquired using an Ettan DIGE Imager (GE HealthCare) at 100 μm resolution and analysed using the Progenesis Samespots analysis software (v4.5; Nonlinear Dynamics). The quality of spot matching was checked and minor corrections were manually applied to improve image alignment. After background subtraction, Cy3 and Cy5 spot intensities were normalized to the intensity of corresponding Cy2 spots. Intensity values were then normalized to those from the ZT5 group, which were given an arbitrary value of one.

After statistical analysis (see below), differential protein spots were excised using a screen picker and one touch spot picker device (Proteomics Consult, Belgium). Four matched spots from four different gels were pooled before in-gel reduction and alkylation of proteins (Massprep Station, Waters, MicroMass, Manchester, UK) as previously reported [47]. Eight μL of a 6.7 ng/L trypsin solution (Promega, Madison, WI, USA) in 25 mM NH_4_HCO_3_ were then added to the gel spots before incubation for 12 h at 37 °C. The resulting peptides were extracted using 20 μL of a 60% acetonitrile (Carlo Erba, Val de Reuil, France) solution containing 0.1% of formic acid. Acetonitrile was removed by vacuum drying using a speedvac prior to nanoLC-MS/MS analyses.

Tryptic peptides were analyzed on a nanoUPLC (NanoAcquityUPLC, Waters, Milford, MA, USA) coupled to a hybrid mass spectrometer, either the MaXis Q-Tof (Bruker Daltonics, Billerica, MA, USA) or Synapt HDMS G1 Q-Tof (Waters, Milford, MA, USA) equipped with a Z-spray ion source and a lock mass system. The chromatographic solvent system consisted of 0.1% HCOOH in water (solvent A) and 0.1% formic acid in acetonitrile (solvent B). Three µL of each sample were concentrated/desalted on a trap column (C18, 180 μm × 20 mm, 5 µm; Waters) at 1% of B at a flow rate of 5 µL/min for 3 min. Afterwards, peptides were eluted from the separation column (BEH130 C18, 75 μm × 250 mm, 1.7 μm; Waters) maintained at 60 °C using a 9 min gradient from 1–35% of B at a flow rate of 450 nL/min.

Mass spectrometers were operated in positive mode, with automatic switching between MS and MS/MS scans. The source temperature of the maXis was set to 160 °C with a spray voltage of −4.5 kV and dry gas flow rate of 5 L/min. Full scan MS spectra were acquired within a mass range of 100–2500 *m*/*z*. External mass calibration of the Tof (MaXis) was achieved before each set of analyses using Tuning Mix (Agilent Technologies, Paolo Alto, CA, USA) in the mass range of 322–2722 *m*/*z*, and mass correction was achieved by recalibration of acquired spectra to the applied lock mass using hexakis (2,2,3,3,-tetrafluoropropoxy)phosphazine ([M + H]+ 922.0098 *m*/*z*). MS acquisition time was set to 0.4 s, and MS/MS acquisition time to a range from 0.05 s (intensity > 250,000) to 1.25 s (intensity < 5000), and ions were excluded after acquisition of two MS/MS spectra with release of exclusion after 1 min. Up to five most intense multiply charged precursors per MS scan were isolated, using an isolation window adapted to the isolated *m*/*z* (2–5 *m*/*z*), then fragmented using energy collisional dissociation. This system was fully controlled by compass HyStar v3.2 and OtofControl v3.2 (Bruker Daltonics, GE).

For the Synapt G1, the capillary voltage was set at 3.5 kV, the sample cone voltage at 35 V, and the extraction cone voltage at 4.0 V. The TOF was calibrated using Glu-fibrino-peptide B on the 50–2000 *m*/*z* range. Online correction of this calibration was performed with Glu-fibrinopeptide B as the lock-mass: the ion (M + 2H)2+ at *m*/*z* 785.8426 was used to calibrate MS data and the ion (M + H)+ at *m*/*z* 684.3469 was used to calibrate MS/MS data. This was done via a lock spray interface, with a lock spray frequency set at 30 s. For tandem MS experiments, automatic switching between MS and MS/MS modes was set as follows: The three most abundant peptides (intensity threshold: 40 counts·sec^−1^), preferably doubly and triply charged ions, were selected on each MS spectrum for further isolation. Collision-induced dissociation (CID) fragmentation using argon as collision gas was performed using two different energies (from *m*/*z* 300 to 500: 14 and 18 eV; from *m*/*z* 501 to 600: 19 and 24 eV; from *m*/*z* 601 to 700: 24 and 28 eV; from *m*/*z* 701 to 800: 28 and 32 eV; from *m*/*z* 801 to 900: 32 and 39 eV; from *m*/*z* 901 to 1000: 39 and 45 eV; from *m*/*z* 1001 to 1200: 45 and 55 eV; from *m*/*z* 1201 to 1700: 55 and 60 eV), which were set using the collision energy profile. For MS and MS/MS acquisitions, the scan range was from *m*/*z* 100 to 2500, with a scan time of 0.5 s. in MS and 0.7 s. in MS/MS mode. The system was fully controlled by the MassLynx software (v.4.1., Waters).

MS/MS data were analysed using the Mascot^TM^ search engine (v2.5.1, Matrix Science, London, UK) installed on a local server. Spectra were searched against a target-decoy version of the *Mus musculus* protein database downloaded from Swissprot (December 2017, 16872 target + decoy entries) to which common contaminants (e.g., trypsin and keratins) were automatically added using the MSDA software suite [48]. Mascot search parameters included MS and MS/MS tolerances of respectively 10 ppm and 0.05 Da (MaXis) or 25 ppm and 0.07 Da (Synapt). A maximum of one missed cleavage was allowed, and carbamidomethylation of cysteine residues was set as fixed modification, and oxidation of methionine residues as variable modification.

Stringent filtering criteria were applied using Proline software (v1.5; http://proline.profiproteomics.fr/) to obtain high-confidence identifications (Mascot peptide ion score > 25; peptide FDR < 1% based on e-values, protein FDR < 1% base on PSM scores, minimum length of seven amino acids). Afterwards, single peptide-based identifications and the identification of common contaminants, such as keratin and trypsin, were not considered. The mass spectrometry proteomics data have been deposited to the ProteomeXchange Consortium via the PRIDE [49] partner repository with the dataset identifier PXD013039.

Among the different proteins that were identified in a given spot, only the major (more abundant) ones were considered to be responsible for variations of spot intensities. The determination of major proteins was performed following a “peptide counting” strategy: The higher the number of peptides assigned to a given protein, the more abundant this protein is. More precisely, and while taking into account the fact that tryptic sites are followed or not by a Proline, the possible missed cleavages and the adequate size of peptides for their detection by mass spectrometry (i.e., peptides of seven to 39 amino acids on the basis of our data), we compared the theoretical number of detectable tryptic peptides to the experimental number we identified. The theoretical number of detectable tryptic peptides was similar for major and minor proteins in given gel spots (mean ratio major/minor = 1.1 ± 0.2), the fraction of detected over theoretical peptides was 3.6 ± 1.8 times greater for major than minor proteins with 4.0 ± 2.7 times more peptides identified for major versus minor proteins (13 ± 8 vs. only 3 ± 1).

Enrichment and functional annotation analysis of transcriptomics data was performed using the desktop version of DAVID (Ease v2.1) and Gene Ontology (GO) databases (March 2018). Enriched GO terms were filtered by only considering those with an Ease score lower than 0.1, a Benjamini *p* value lower than 0.05 and a fold enrichment higher than 2. Enriched GO terms were grouped together into broad functional categories, which were then considered as enriched broad functions.

### 4.4. Western-Blot Analyses

The second half of each cerebellum was grinded under liquid nitrogen for 30 s at 30 Hz using a Mixer Mill MM400 (Retsch, Eragny sur Oise, France), and proteins were extracted from the resulting powder using 10 µL of extraction buffer per mg of tissue. Extraction buffer was composed of 8 M urea, 2 M thiourea, 2% Chaps, 10 mM DTT, 30 mM Tris pH8.8 and protease inhibitors (ThermoFisher). After sonication (10 s at 135 watts) and incubation at 4 °C under agitation for 90 min, cell debris were pelleted by centrifugation and proteins were TCA-precipitated at 4 °C. After centrifugation (15,000× *g*, 15 min, 4 °C), protein pellets were washed with cold acetone, and then dissolved in Laemmli buffer. Total protein concentrations were determined using the Bradford Protein Assay kit from Bio-Rad.

Five μg of each sample were separated on 8–16% Bio-Rad Mini-Protean TGX Stain-Free Precast Gels (Bio-Rad). Gels were imaged after activation using the Bio-Rad ChemiDoc Touch Imaging System and transferred to nitrocellulose membranes (0.2 μm) using the Bio-Rad Trans-Blot Turbo Transfer System. Blots were immediately imaged to check for proper transfer, they were blocked for 1 h at room temperature with a solution of TBS-T (Tris 25 mM, NaCl 137 mM, KCl 2.68 mM, 5% Tween 20) containing 4% of BSA, and they were then incubated overnight at 4 °C with primary antibodies targeting the regulation of two proteins highlighted by proteomics analysis, Vesicle-fusing ATPase (NSF; sc-74458, Santa Cruz Biotechnology, Dallas, TX, USA) and Synapsin-2 (SYN2; ab13258, Abcam, Paris, France). Primary antibodies were used at 1/500 (NSF) and 1/5000 (SYN2) dilution in the blocking solution. After three washings for 10 min each in TBS-T, blots were incubated for 1 h at room temperature with an anti-rabbit (sc-2004) or anti-mouse (sc-2005) peroxidase-conjugated secondary antibody from Santa Cruz Biotechnology diluted (1/5000) in the blocking solution. After three washings for 10 min each in TBS-T, blots were incubated for 1 min in Luminata Classico Western HRP substrate (Merck Millipore, Molsheim, France) and then imaged for chemiluminescence using the ChemiDoc Touch Imaging System (Bio-Rad). Image analysis was done using the Bio-Rad Image Lab software v5.1. Signals were normalized to total proteins, as measured on the stain-free gel image, and intensity values were expressed relative to those from ZT5 mice, which were assigned an arbitrary value of one.

### 4.5. RNA Extraction and Real-Time qPCR Analysis

Pieces of frozen cerebellum from *Per1^-/-^/Per2^Brdm1^* and WT control mice were homogenized in phenol/guanidine-based reagent and total RNA was extracted according to the manufacturer’s protocol (RNeasy^®^ Lipid Tissue mini kit, Qiagen France SAS, Courtaboeuf). The concentration and purity of RNA were evaluated on NanoDrop ND-100 spectrophotometer (NanoDrop Technologies, Wilmington, DE, USA; A260/A280, and A260/A230 values were > 1.8). Integrity of the RNA was assessed using the 2100 Bioanalyzer (Agilent Technologies, Santa Clara, CA; RIN > 7). cDNAs were synthesized from 100 ng of total RNA using the Fluidigm Reverse Transcription Master Mix (Fluidigm P/N-100-6297). A specific pre-amplification (STA: Specific Target Amplification) was performed in a multiplex Polymerase Chain Reaction (PCR) using a pool of all primers in each cDNA sample and a pool of cDNA (Fluidigm P/N 100-5744).

qPCR step was performed on 48.48 Dynamic Array™ integrated fluidic circuit (IFC; Fluidigm) by the GenomEast platform, a member of the “France Génomique” consortium (ANR-10-INBS-0009). No-template reactions were performed as negative controls for each primer pair. The PCR mix contained the internal passive reference dye 6-carboxyl-X-rhodamine (ROX) for normalization of the potential non-PCR-related fluorescence fluctuations. A dilution curve of pooled cDNA samples was used to calculate the amplification efficiency for each primer set and determine the optimal cDNA dilution according to the manufacturer’s instructions. Relative expression levels were determined using the comparative ΔCT method to normalize target gene mRNA to *beta-2-microglobilin* (*B2m*) (reference below).

The following TaqMan Gene Expression Assays (Applied Biosystems, Foster city, CA, USA) were used:

*B2m* (Mm00437762-m1), *Bmal1* (*Arnt1*, Mm00500226_m1), *Nsf* (Mm00435390-m1), *Sncb* (Mm00504325-m1), *Syn2* (Mm01184324-m1), *Fabp7* (Mm00445225-m1) and *Tpi1* (Mm00833691-g1).

### 4.6. Statistical Analysis

Statistical analysis of data from quantitative proteomics, western-blot and qPCR was performed using the package ‘cosinor’ under R software environment v3.4.0 [50]. We used the following regression model: y = A + (B·cos(2π·(x − C)/24)), where A is the mesor (midline statistic of rhythm, a rhythm-adjusted mean), B the amplitude (corresponding to half the extent of predictable variation within a cycle) and C the acrophase (i.e., the time of highest values in each cycle). For 2D-DIGE, protein spots exhibiting significant cosinor *p* values (*p* < 0.05) for the mesor, amplitude, and acrophase were considered as rhythmic. Data are presented as means ± standard error of the mean (SEM).

## 5. Conclusions

This study reveals the widely rhythmic feature of the proteome of the cerebellum. Further studies will be necessary to define whether such daily accumulation of proteins depends specifically on the intrinsic cerebellar clock, and/or on rhythmic behaviors such as sleep/wake cycle and feeding/fasting cycle orchestrated by the master clock in the suprachiasmatic nuclei.

In addition, it will be important to determine the characteristics of circadian rhythmicity of proteins according to the various neuronal and glial cell types present in the cerebellar network.

## Figures and Tables

**Figure 1 ijms-20-01852-f001:**
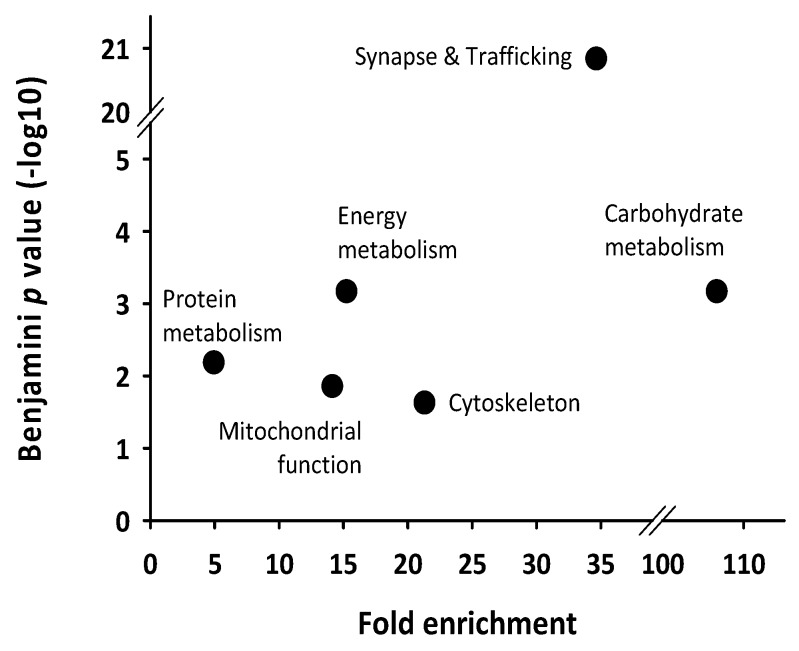
Functional annotation analysis from the set of rhythmically expressed proteins according to cosinor regression. The broad functions affected over the circadian day in the mice cerebellum were highlighted by setting thresholds for the fold enrichment (>2) and statistical analysis (Ease score < 0.1 and Benjamini *p* value < 0.05) that were computed using the scripts contained in the desktop version of DAVID (Ease v2.1).

**Figure 2 ijms-20-01852-f002:**
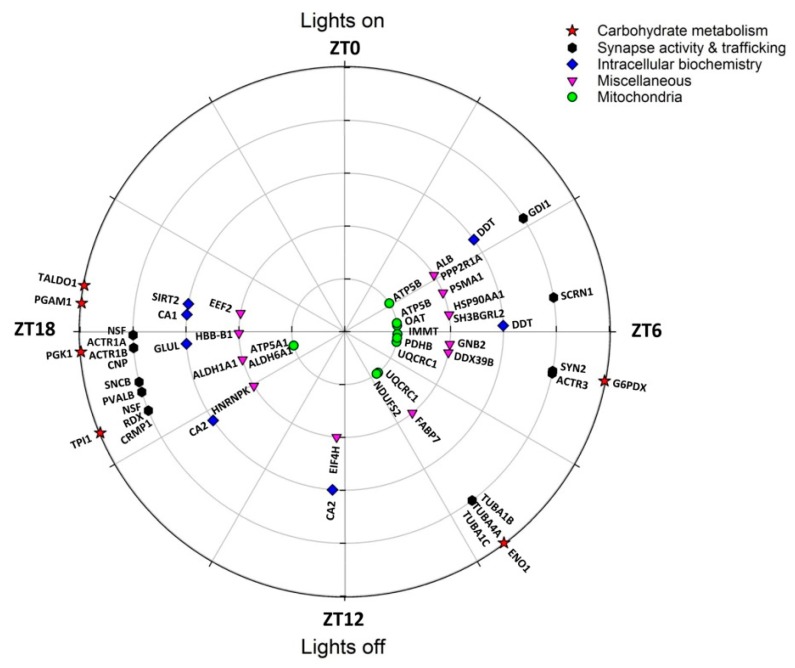
Polar plot showing the temporal distribution of peak phase of rhythmic proteins in the mouse cerebellum according to their function.

**Figure 3 ijms-20-01852-f003:**
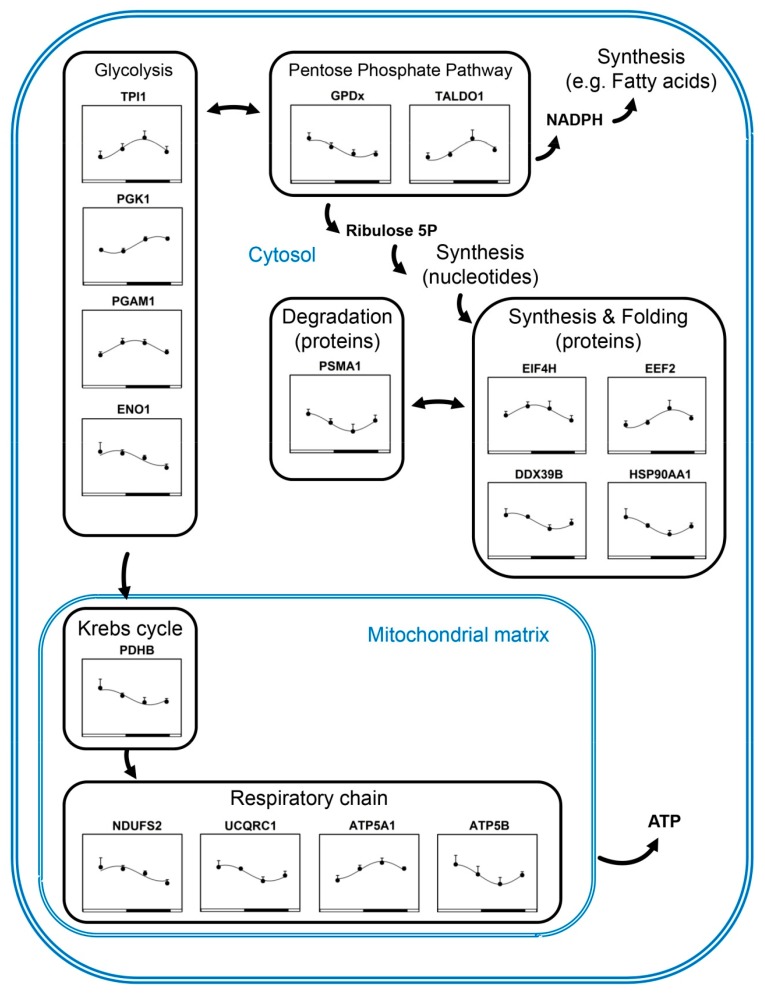
Visualisation of the daily expression of proteins on metabolic pathways in the cerebellum. Fitted curves represent significant cosinor regressions. See also Table 1 and Appendix A. Data are shown as mean ± SEM.

**Figure 4 ijms-20-01852-f004:**
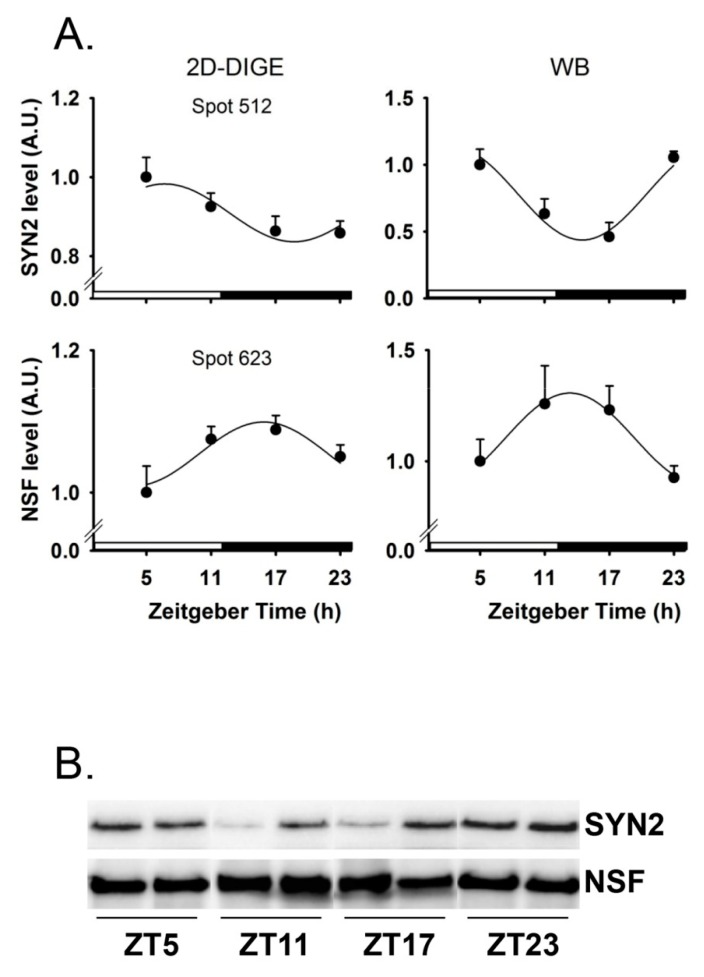
(**A**) Daily expression of Synapsin 2 (SYN2) and Vesicle-fusing ATPase (NSF) in the cerebellum, assessed by two-dimensional difference in gel electrophoresis (2D-DIGE; left panels; n = 4–6 per ZT) and western blots (WB; right panels; *n* = 5 per ZT). Fitted curves represent significant cosinor regressions. A.U., arbitrary unit. Data are presented as mean ± SEM. (**B**) Representative western blots for SYN2 and NSF. ZT, Zeitgeber time.

**Figure 5 ijms-20-01852-f005:**
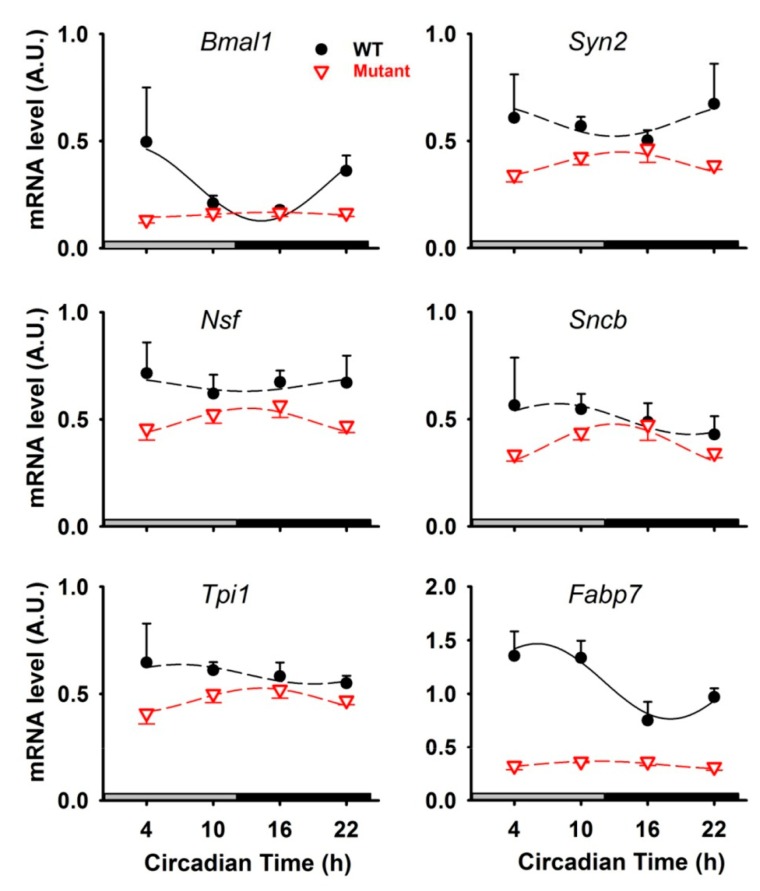
Circadian expression of genes in the cerebellum of wild-type (WT, black curves) and *Per1^-/-^/Per2^Brdm1^* mutant mice (Mutant, red curves). Fitted solid and dashed curves represent significant and non-significant cosinor regressions, respectively. Data are presented as means ± SEM. A.U., arbitrary unit. *Bmal1*, Brain and Muscle Aryl-hydrocarbon receptor nuclear translocator-like protein 1; *Syn2*, Synapsin 2; *Nsf*, Vesicle-fusing ATPase; *Sncb*, β-synuclein; *Tpi1*, Triosephosphate isomerase 1; *Fabp7*, Fatty acid binding protein 7.

**Table 1 ijms-20-01852-t001:** Daily variations of abundant proteins in the mouse cerebellum.

Spot N°	Protein Name	Gene Name	Subcellular Location	Related Functions	Peak Phase (ZT)
512	Glucose-6-phosphate 1-dehydrogenase X	*G6pdx*	Cytosol, nucleus	Carbohydrate metab	6.7
1056	α-enolase	*Eno1*	Cell membrane, cytoplasm	Carbohydrate metab	9.5
167	Triosephosphate isomerase	*Tpi1*	Cytosol	Carbohydrate metab	16.5
382	Phosphoglycerate kinase 1	*Pgk1*	Cytoplasm	Carbohydrate metab	17.7
198	Phosphoglycerate mutase 1	*Pgam1*	Cytosol, nucleus	Carbohydrate metab	18.4
315	Transaldolase	*Taldo1*	Cytoplasm	Carbohydrate metab	18.7
540, 1224	ATP synthase subunit β	*Atp5b*	Mitochondrion	Mitochondrial function	3.8, 5.4
401	Ornithine aminotransferase	*Oat*	Mitochondrion	Mitochondrial function	5.6
677	MICOS complex subunit Mic60	*Immt*	Mitochondrion	Mitochondrial function	6.2
1060	Pyruvate dehyd. E1 component subunit β ^1^	*Pdhb*	Mitochondrion	Mitochondrial function	6.5
412, 417	Cytochrome b-c1 complex subunit 1	*Uqcrc1*	Mitochondrion	Mitochondrial function	6.8, 9.4
1056	NADH dehyd. [ubiquinone] iron-sulfur prot 2	*Ndufs2*	Mitochondrion	Mitochondrial function	9.5
1330	ATP synthase subunit α	*Atp5a1*	Mitochondrion	Mitochondrial function	17.0
1330	Methylmalonate-semialdehyde dehydrogenase	*Aldh6a1*	Mitochondrion	Mitochondrial function	17.0
219	Proteasome subunit α type-1	*Psma1*	Cytoplasm, nucleus	Protein metabolism	4.6
1224	Heat shock protein HSP 90-α	*Hspaa1*	Membrane, cytoplasm	Protein metabolism	5.4
412	Spliceosome RNA helicase Ddx39b	*Ddx39b*	Cytoplasm, nucleus	Protein metabolism	6.8
216	Eukaryotic translation initiation factor 4H	*Eif4h*	Cytoplasm (perinuclear)	Protein metabolism	12.3
554	Heterogeneous nuclear ribonucleoprotein K	*Hnrnpk*	Cytoplasm, nucleus	Protein metabolism	15.9
315	Elongation factor 2	*Eef2*	Cytoplasm	Protein metabolism	18.7
540	Rab GDP dissociation inhibitor α	*Gdi1*	Cytoskeleton, Golgi	Synapse & trafficking	3.8
1224	Secernin-1	*Scrn1*	Cytoplasm	Synapse & trafficking	5.4
512	Synapsin-2	*Syn2*	Cytoskeleton, nucleus	Synapse & trafficking	6.7
412	Actin-related protein 3	*Actr3*	Cytoskeleton	Synapse & trafficking	6.8
1056	Tubulin α-1B chain	*Tuba1b*	Cytoskeleton	Synapse & trafficking	9.5
1056	Tubulin α-4A	*Tuba4a*	Cytoskeleton	Synapse & trafficking	9.5
1056	Tubulin α-1C chain	*Tuba1c*	Cytoskeleton	Synapse & trafficking	9.5
630, 623	Vesicle-fusing ATPase	*Nsf*	Cytoplasm	Synapse & trafficking	16.5, 17.9
630	Radixin	*Rdx*	Membrane, cytoskeleton	Synapse & trafficking	16.5
630	Dihydropyrimidinase-related protein 1	*Crmp1*	Cytoplasm, cytoskeleton	Synapse & trafficking	16.5
36	Parvalbumin α	*Pvalb*	Cytoplasm, nucleus	Synapse & trafficking	16.9
82	β-synuclein	*Sncb*	Cytoplasm, synapse	Synapse & trafficking	17.1
382	α -centractin	*Actr1a*	Cytoskeleton	Synapse & trafficking	17.7
382	β-centractin	*Actr1b*	Cytoskeleton	Synapse & trafficking	17.7
382	2′,3′-cyclic-nucleotide 3′-phosphodiesterase	*Cnp*	Cytoskeleton	Synapse & trafficking	17.7
21, 22	D-dopachrome decarboxylase	*Ddt*	Cytoplasm	Miscellaneous	3.6, 5.9
540	PP2A subunit A isoform PR65-alpha	*Ppp2r1a*	Membrane, cytoplasm	Miscellaneous	3.8
582	Serum albumin	*Alb*	[Extracellular, secreted]	Miscellaneous	3.5
22	SH3 domain-binding glutamic acid-rich-like prot 2	*Sh3bgrl2*	Nucleus	Miscellaneous	5.9
1060	Guanine nucleotide-binding protein subu β-2	*Gnb2*	Cytoplasm (perinuclear)	Miscellaneous	6.5
216, 212	Carbonic anhydrase 2	*Ca2*	Membrane, cytoplasm	Miscellaneous	12.3, 15.7
1330	Retinal dehydrogenase 1	*Aldh1a1*	Cytoplasm	Miscellaneous	17.0
382	Glutamine synthetase	*Glul*	Cytoplasm (mainly)	Miscellaneous	17.7
1019	Hemoglobin subunit β-1	*Hbb-b1*	[Red blood cells]	Miscellaneous	17.9
198	Carbonic anhydrase 1	*Ca1*	Cytoplasm	Miscellaneous	18.4
315	NAD-dependent protein deacetylase sirtuin-2	*Sirt2*	Cytoplasm (mainly)	Miscellaneous	18.7
42	Fatty acid-binding protein, brain ^2^	*Fabp7*	Cytoplasm	Lipid transp., PPAR signal.	(9.3)

Subcellular location is based on annotations from the UnitProtKb database. Shaded area indicates nocturnal peak phase. ^1^ Pyruvate dehydrogenase E1 component subunit β (PDHB; Spot 1060) is also involved in carbohydrate metabolism as it is involved in fueling of the Krebs cycle with glycolytic intermediates. ^2^ For fatty acid-binding protein (FABP7; Spot 42), the abundance rhythm was close to, but beyond the threshold of significance set at *p* < 0.05 (*p* value for amplitude = 0.06; see Appendix A for details).

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
