# Peer review of "Circadian Analysis of the Mouse Cerebellum Proteome"

_ijms, 2019, doi:10.3390/ijms20081852_

Round 1
Reviewer 1 Report
Authors should discuss how differences of circadian expression in cerebellum and SCN. It is not clear.
Author Response
We thank Referee 1 for having evaluated our manuscript positively.
Comments and Suggestions for Authors
Authors should discuss how differences of circadian expression in cerebellum and SCN. It is not clear.
Reply:
To better describe the timing difference of the circadian clockwork between the cerebellum and the suprachiasmatic nuclei, first, we have reworded the introduction as follows:
Lines 55-60: “Daily patterns of clock gene expression in the cerebellum and other secondary clocks such as the liver show a temporal organization of the clockwork very close to that in the suprachiasmatic clock, as described above [9,10]. In terms of phasing of the whole clockwork, however, circadian oscillations of clock genes and proteins in the mouse cerebellum are rather similar to those in the murine liver, the oscillations in both structures being phase-delayed by about 6 h as compared to those in the master clock. For instance, Per1 and Dbp mRNA profiles peak around dusk (corresponding to activity onset in mice) in both the cerebellum and liver, while these profiles display peaks around midday (mid-resting period) in the suprachiasmatic nuclei (i.e., 6 h earlier)[11-13].”
Second, at the beginning of the discussion, we have completed the comparison of circadian expression of proteins between the cerebellum, the liver and the suprachiasmatic nuclei, as follows:
Lines 196-210: “Amongst these proteins, there were two main peaks of abundance, centered either in the middle of the rest (midday) or active phases (midnight). Of note, such bimodal clustering of rhythmic cerebellar proteins over 24 h differs from the suprachiasmatic nuclei in which the peaks of protein abundance are more widely distributed in phase across the 24 h [17,18]. Thus, the daily distribution of rhythmic proteins in the cerebellum did not match the lagged temporal organization of its clockwork when compared to the suprachiasmatic nuclei. These findings therefore suggest complex temporal regulations that differ between cerebral structures. In the peripheral clock of the liver, circadian profiles of protein expression are also differently distributed over 24 h because in that case, they fall into three distinct phase clusters, the main one peaking at night and two minor ones peaking either around midday or at the day-to-night transition [16].”
Reviewer 2 Report
The present study asses the circadian analysis of the mouse cerebellum proteome. The daily oscillations of cerebellar proteins were investigated in mice using large-scale two-dimensional differences in gel electrophoresis (2D-DIGE)
Introduction provides sufficient background and includes relevant references. The research design is appropriate, the results are clearly presented and the conclusions are enough supported by the results. English language and style are sufficiently clear and understandable
I think that Originality, Significance of Content, Quality of Presentation and Overall Merit meet the requirements of the journal. The content is interesting to the readers
My recommendation is to accept after minor revision
3. Discussion
The authors should discuss the data related to the mutant mice
5. Materials and methods
5.2 Even if the results are not affected, the authors should explain why they used different ZT for the wild type mice respect the CT used for mutant mice.
In Figure 5 when they compare wild type vs mutant mice, the CT is referred to the mutants. The consequence is 1 hour difference but perhaps that could explain the ±SEM you have in CT4 (ZT5) Bmal
Table 1: Daily variations of abundant proteins in the mouse cerebellum
Figure 2. Polar plot showing the temporal distribution of peak phase of rhythmic proteins in the mouse cerebellum according to their function.
Author Response
We thank Referee 2 for having evaluated our manuscript positively.
3. Discussion
The authors should discuss the data related to the mutant mice
Reply:
To take into account Referee 2’s comment, we have added the following sentences in the discussion:
Lines 213-217: “(…) the present study shows that the clock gene Bmal1 and the clock-controlled gene Fabp7 are rhythmically expressed in the cerebellum of wild-type mice, while these rhythms are totally abolished in mice with defective clocks (i.e., Per1-/-/Per2Brdm1 double mutant mice). Such observation confirms that the cerebellum activity is controlled, at least in part, by a circadian clock.”
5. Materials and methods
5.2 Even if the results are not affected, the authors should explain why they used different ZT for the wild type mice respect the CT used for mutant mice.
Reply:
To answer to this point, we added new sentences as follows:
Lines 371-376: “Note that the timing of cerebella sampling in these control and mutant mice was phase-advanced by 1 h, as compared to the previous experiment in wild-type C57BL/6J mice. Because we assessed circadian phase by non-linear sine regression (cosinor) over the 4 time-points (see statistical analysis for details) and we did not compare data between the two experiments, the 1-h difference in sampling does not affect the validity of our results.”
In Figure 5 when they compare wild type vs mutant mice, the CT is referred to the mutants. The consequence is 1 hour difference but perhaps that could explain the ±SEM you have in CT4 (ZT5) Bmal
Reply:
As explained in the material & methods section, the wild-type mice of the qPCR experiment (genetic background: C57BL6 X 129Sv; experiment 2) have been sampled exactly at the same time as double mutant mice, that is, starting at CT4. Thus, CT time-points refer to both control and mutant mice.
Other wild-type mice (genetic background: C57BL6) have been sampled for the proteomic analysis (experiment 1), starting at ZT5.
Table 1: Daily variations of abundant proteins in the mouse cerebellum
Reply:
Modified accordingly.
Figure 2. Polar plot showing the temporal distribution of peak phase of rhythmic proteins in the mouse cerebellum according to their function.
Reply:
Modified accordingly.